# FTP: A Fine-grained Token-wise Pruner for Large Language Models via Token Routing

## Abstract

Recently, large language models (LLMs) have demonstrated superior performance across various tasks by adhering to scaling laws, which significantly increase model size. However, the huge computation overhead during inference hinders the deployment in industrial applications. Many works leverage traditional compression approaches to boost model inference, but these always introduce additional training costs to restore the performance and the pruning results typically show noticeable performance drops compared to the original model when aiming for a specific level of acceleration. To address these issues, we propose a fine-grained token-wise pruning approach for the LLMs, which presents a learnable router to adaptively identify the less important tokens and skip them across model blocks to reduce computational cost during inference. To construct the router efficiently, we present a search-based sparsity scheduler for pruning sparsity allocation, a trainable router combined with our proposed four low-dimensional factors as input and three proposed losses. We conduct extensive experiments across different benchmarks on different LLMs to demonstrate the superiority of our method. Our approach achieves state-of-the-art (SOTA) pruning results, surpassing other existing pruning methods. For instance, our method outperforms BlockPruner and ShortGPT by approximately 10 points on both LLaMA2-7B and Qwen1.5-7B in accuracy retention at comparable token sparsity levels.

## 1 Introduction

Recently, large language models (Zhao et al., 2023; Minaee et al., 2024) draw much attention to various natural language process (NLP) tasks due to their superiority, which mainly benefits from the great success of the ChatGPT series. Now the LLMs design usually follows the scaling law to make the models huger and more complex to improve the performance of the LLMs, which leads to substantial memory usage and computational demands. However, these would hinder the deployment of LLMs in industrial applications, even if they have outstanding capability in various tasks. Therefore, many works are proposed to boost the LLMs inference while maintaining accuracy, which include model pruning (Gao et al., 2020; Li et al., 2023a; Wang et al., 2024), quantization (Dettmers et al., 2024; Yao et al., 2022), knowledge distillation (Huang et al., 2022; Gu et al., 2024) and conditional computing technique (Schuster et al., 2022; Liu et al., 2023; Akhauri et al., 2024).

Quantization technique usually quantizes the float weights and activation values into low-bit representation to accelerate the kernel computation. Knowledge distillation usually leverages large teacher model to guide the small student model learning the prediction distribution from the teacher model, which can improve the small model performance for deployment. And conditional computing technique usually dynamically activates the weights or activations of the model instead of directly removing them. Model pruning is a popular technique in industrial applications to boost model inference, which usually identifies and removes the less important weights to reduce the computation overhead. Model pruning methods can be broadly categorized into two classes, which are structured pruning and unstructured pruning. Structured pruning is preferred over unstructured pruning since it does not require the specific acceleration hardware or software library for speedup in deployment. Many works (Zhao et al., 2024; Ma et al., 2023) adopt the traditional compression technique to prune the LLMs for acceleration, which requires retraining the LLMs to restore accuracy. However, the retraining process requires computing overhead, which is inefficient for deployment in applications. Recently, some works (Men et al., 2024; Zhong et al., 2024) find much redundancy in

depth of the LLMs, and pruning in depth achieves better results than pruning in width. However, we argue that block removal is a coarse-grained pruning approach, which cannot exploit the potential of LLM pruning.

To address these issues, we present a fine-grained token-wise pruning method for the LLMs, which doesn't need to retrain the LLMs while tremendously maintaining the accuracies of the LLMs. Firstly, we make a deep analysis of the token redundancy across different blocks in the LLMs, proving there is much room for fine-grained token-wise pruning. Then, we propose a token-wise pruning framework for LLMs, which incorporates a sparsity scheduler to allocate a sparsity ratio for each block and a dynamic router to prune unimportant tokens in the sequence, based on four key factors. Initially, we introduce an efficient sparsity search strategy with a static router to construct the sparsity scheduler. Using the searched static router as a starting point, we then train our dynamic router. Instead of relying directly on hidden states, we propose four low-dimensional factors as input to the router, making the model easier to train. Additionally, we introduce three losses—guide loss, sparsity constraint loss, and distillation loss—to fine-tune the dynamic router. Finally, we re-search the sparsity scheduler with a trained router to refine the sparsity configuration. These innovative components contribute to a robust and effective token-wise LLM pruning method.

To verify the effectiveness of our method, we conduct extensive experiments on various LLMs including Qwen and LLaMA series models with our token-wise pruning method. And our method significantly surpasses the other SOTA pruning methods by a large margin without retraining the LLMs, which fully demonstrates the superiority of our method. In a summary, our contributions can be mainly summarized as follows:

- We have been diving into an analysis of the deep redundancy of token features across different blocks of the LLMs, proving that there is much room for token-wise pruning in LLMs.
- We present a token-wise pruning framework consisting of three main steps: initial sparsity search using a static router for sparsity allocation, dynamic router training with our proposed four factors and three losses, and sparsity scheduler fine-tuning with the trained router.
- Extensive experiments have been conducted on various LLMs with our proposed method, which indicates our method surpasses other SOTA pruning methods for LLMs by a large margin. These results demonstrate the superiority of our proposed token-wise pruning approach.

## 2 RELATED WORK

**LLM Pruning.** Pruning techniques in LLMs aim to identify and remove redundant weights or tokens from models. These methods aim to decrease computational complexity, inference overhead and memory usage by efficiently ignoring pruned elements during computation. Given the recent exponential increase in model size, significant research has been dedicated to optimizing LLM inference. As for weight-level pruning, SparseGPT (Frantar & Alistarh, 2023) addresses the layer-wise reconstruction problem for pruning by computing Hessian inverses. Wanda (Sun et al., 2023) introduces a pruning criterion that involves multiplying weight magnitudes by input feature norms. Moreover, FLAP (An et al., 2024), LLM-Pruner (Ma et al., 2023), Sheared-LLaMA (Xia et al., 2023) and BlockPruner (Zhong et al., 2024) eliminate coupled structures in the aspect of network width while retaining the number of layers, while FoldGPT (Liu et al., 2024) and ShortGPT (Men et al., 2024) exploit model depth redundancy to obtain lightweight models. As for token-level pruning, Selective Context (Li et al., 2023b) proposes to merge tokens into units, and then applies prompt pruning based on the self-information indicator. STDC (Yin et al., 2023) prunes the prompts based on the parse tree, which iteratively removes phrase nodes that cause the smallest performance drop after pruning it. LLMLingua (Jiang et al., 2023a) and LongLLMLingua (Jiang et al., 2023b) perform demonstration-level pruning followed by token-level pruning based on perplexity. PCRL (Jung & Kim, 2024) introduces a token-level pruning scheme based on reinforcement learning. However, most existing pruning approaches permanently remove weights or tokens, which may significantly degrade accuracy for more challenging tasks. In this work, we present a fine-grained token-wise pruner which can adaptively prune tokens within each block of LLMs based on the varying inputs via token routing during inference.

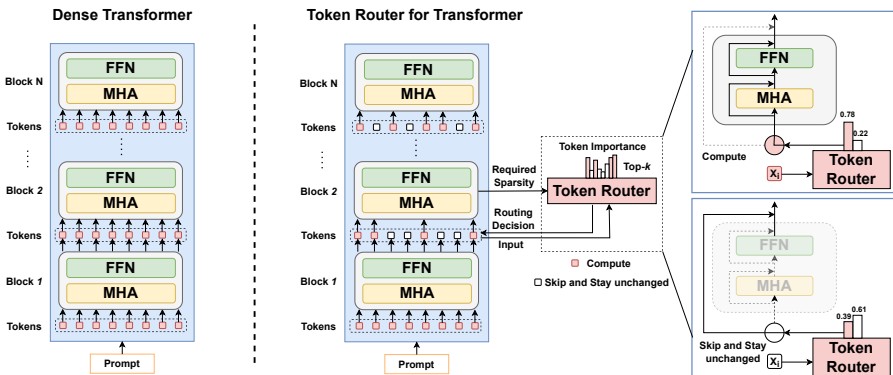

Figure 1: Overview of LLM structure and router workflow. (Left) Dense Transformer where all tokens are processed in every block. (Middle) Token Router for Transformer, which dynamically selects tokens to compute or skip based on their importance and block-wise sparsity at each block. (Right) A detailed view of how the Token Router uses token importance features to make binary decisions (compute or skip) for each token within a block.

**Conditional Computing.** Static pruning permanently removes weights or tokens from LLMs, which can result in a significant drop in accuracy, especially for more challenging tasks. A wide variety of recent work has developed to dynamically activate weights or token instead of removing them, also named as conditional computing. DejaVu (Liu et al., 2023) dynamically activates neurons and attention heads of each LLM's layer by building predictors to estimate sparsity patterns. ShadowLLM (Akhauri et al., 2024) dynamical activates weights based on the context (input) itself by training a predictor to predict the sparsity pattern dependent on the input tokens. However, the sparse activation of weights still hurts the generability of models. Many works (Elbayad et al., 2020; Liu et al., 2021; Schuster et al., 2022) utilize early exiting to learn to decide when to end computation on a given token, allowing the token to skip any remaining transformer layers. MoD (Raposo et al., 2024) dynamically selects tokens via a trainable router for each block which takes hidden states as input and manually specifies the sparsity ratios for every block, and requires training from scratch. In contrast, our work proposes a global token router that takes designed input instead of hidden states, combined with a sparsity scheduler using a static router for pruning sparsity allocation for all blocks. It is trained to evaluate token importance to control tokens' skipping or computation for each block without the requirement for LLM's retraining.

## 3 METHOD

In this study, we present a fine-grained token-wise pruner (FTP) for large language models (LLMs) via token routing, which leverages a simple yet effective neural network to predict less important tokens to skip during inference in each transformer block. The primary goal is to reduce token redundancy along the depth dimension by selectively skipping token computation in each transformer block, thereby significantly accelerating LLMs while minimizing performance degradation. As illustrated in Figure 3, our FTP consists of three main steps: initial sparsity search, dynamic router training, and sparsity scheduler fine-tuning. First, we employ a genetic algorithm (GA) to search for block-wise sparsity scheduler using a proposed static router. Then, we fix the scheduler and train the dynamic router with three proposed loss functions. Finally, the router is frozen, and the scheduler is fine-tuned again. In this section, we first review the structure of LLMs. Next, we provide a detailed analysis of token redundancy along the depth dimension of LLMs in Section 3.1, highlighting the potential for token-wise pruning in LLMs. A comprehensive explanation of our method is presented in Section 3.2.

### 3.1 PRELIMINARY

**LLM Structure.** The mainstream large language models (LLMs) are mostly built upon the transformer architecture, which heavily relies on attention mechanisms. The mechanisms allow the model to attend to different parts of an input sequence, making it highly effective for sequence-to-sequence

tasks. A typical transformer consists of several identical blocks, each refining the input data through a combination of attention and feed-forward mechanisms. Each transformer block consists of two main components: multi-head attention (MHA) and feed-forward network (FFN) as depicted in Figure 1. The attention mechanism is applied multiple times in parallel among the token sequence, allowing the model to focus on different parts of the sequence at different positions. The attention computation is calculated as:

$$\text{Attention}(Q, K, V) = \text{softmax}\left(\frac{QK^T}{\sqrt{d}}\right) V \tag{1}$$

where $Q, K, V \in \mathbb{R}^{L \times d}$ are the query, key, and value matrices of the input sequence, $L$ is the sequence length, and $d$ is the dimension of the key. The result of the attention mechanism for each token is a weighted sum of all the tokens in the sequence. After applying attention, the model passes the output through a feed-forward network that consists of two fully connected layers with a non-linear layer. The forward computation of the $i$-th block in a transformer can be expressed as follows:

$$\begin{aligned} X_i' &= \text{MHA}(\text{LN}(X_i)) + X_i \\ X_{i+1} &= \text{FFN}(\text{LN}(X_i')) + X_i' \end{aligned} \tag{2}$$

where $X_i \in \mathbb{R}^{L \times d}$ is the input of the $i$-th block, LN is layer normalization applied to the inputs, MHA is the multi-head attention mechanism, and FFN is the feed-forward network.

The computation complexity of a transformer block is mostly dominated by two components: the MHA and the FFN. The MHA has a complexity of $O(L^2 \cdot d)$, where $L$ is the sequence length and $d$ is the hidden dimension, due to the pairwise attention computation across tokens. The FFN, which processes each token independently, has a complexity of $O(L \cdot d^2)$. Therefore, the overall complexity of a transformer block is $O(L^2 \cdot d + L \cdot d^2)$, where the quadratic dependency on $L$ makes attention particularly expensive for long sequences (Clark et al., 2020).

**Token Redundancy.** Ideally, transformers could optimize their computational budget by allocating resources more effectively and avoiding unnecessary computation. Previous works have shown that transformers exhibit certain semantic capabilities in earlier blocks (Hasan et al., 2021), and there is substantial block-wise redundancy throughout the model (Men et al., 2024). Additionally, previous work (Raposo et al., 2024) demonstrates that selectively dropping tokens across blocks can still maintain performance comparable to a fully dense transformer. In this work, we uncover significant token-wise redundancy across blocks during the inference phase of pretrained transformers.

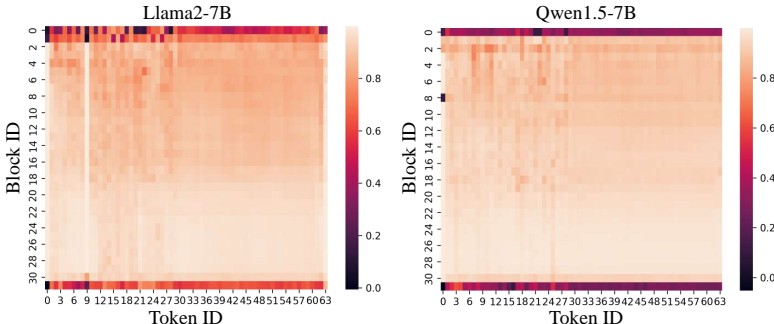

Figure 2: Token similarity across different transformer blocks.

As illustrated in Figure 2, we have randomly selected 50 sequences from the training dataset, each consisting of 64 tokens, and calculated the similarity between the input hidden state and output hidden state from each token of all blocks on both the LLaMA2-7B-base and Qwen1.5-7B-base models. Higher similarity indicates that a block has less influence on the token, while greater changes in hidden states suggest lower token redundancy. Our analysis reveals the following key insights:

First, we observe substantial token redundancy across both models. Specifically, 89.94% and 93.16% of tokens in LLaMA2-7B and Qwen1.5-7B, respectively, exhibit a similarity score higher than 0.8, suggesting minimal changes and a high potential for pruning. Conversely, only 10.06% and 6.84% of tokens have similarity scores below 0.8, indicating meaningful transformations.

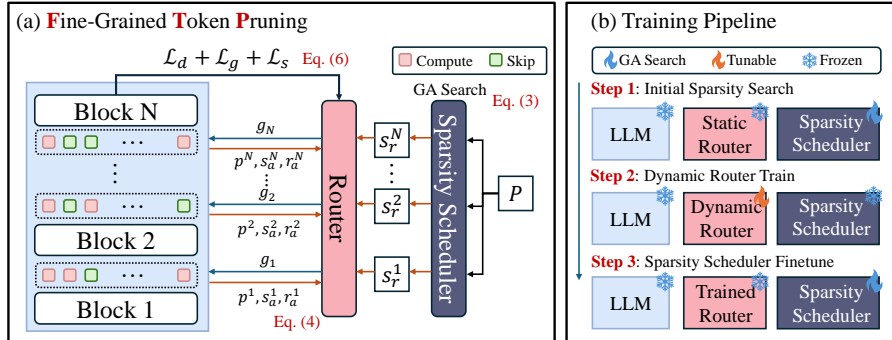

Figure 3: Overview of our method. (a) Our Fine-Grained Token Pruning uses token position $p$, absolute attention scores $s_a$, relative attention score rank $r_a$ and sparsity requirement $s_r$ to guide gate prediction, skipping computation instead of discarding tokens. A GA-based scheduler optimizes sparsity per block, and the router is trained with three proposed losses. (b) We decouple sparsity scheduling and router training into three steps, simplifying the optimization.

Second, token redundancy varies across the blocks of the transformer. Tokens in the initial and final blocks show more significant changes, while tokens in the middle blocks exhibit greater redundancy. Specifically, in the first and last three blocks, 49.74% and 35.42% of tokens have similarity scores below 0.8, while in the middle blocks, 99.10% and 99.76% of tokens have similarity scores above 0.8 in LLaMA2-7B and Qwen1.5-7B, respectively.

**Token Router in LLMs.** Transformers capture contextual information and predict the next token by leveraging the effectiveness of the attention mechanism. However, the computational cost of large language models (LLMs) is extremely high. As discussed above, the attention mechanism in a transformer block has a complexity of $O(L^2 \cdot d + L \cdot d^2)$, meaning the FLOPs of an LLM grow exponentially with the number of tokens. In this context, token routing, which selectively allows only a subset of tokens to participate in each transformer block's computation, presents an effective approach to reduce the sequence length processed by each block, significantly decreasing the overall computation cost.

Figure 1 illustrates the mechanism of a typical token router (Raposo et al., 2024). For each block, the hidden states of input tokens are assessed by the token router, which predicts the importance of each token. A specific proportion of the most important tokens is then selected to undergo the block computation based on sparsity requirements, including multi-head attention and MLP layers. The unselected tokens, meanwhile, skip the block's computation and remain unchanged until next block.

**Simultaneously Allocating Sparsity and Pruning Tokens Is Non-Trivial.** As shown in Figure 2, we observe that the redundancy levels of different blocks are inconsistent, and the redundancy patterns of tokens within the same block are not fixed. Therefore, we need to design a scheme that can simultaneously determine the sparsity ratios for each block and the pruning patterns for each block's tokens. However, optimizing both the sparsity allocation and token pruning patterns across blocks increases the complexity of the optimization. Previous methods typically relied on empirical values to manually specify sparsity rates for each block, which can result in suboptimal performance.

### 3.2 FINE-GRAINED TOKEN-WISE PRUNER

We design a fine-grained token-wise pruner that utilizes a dynamic router to control which tokens should be computed or skipped within a block during the forward process. To address the challenge of simultaneously allocating sparsity and optimizing tokens, we divide the problem into several steps. As shown in Figure 3, our pruning pipeline consists of three main steps: 1) initial searching for a sparsity scheduler; 2) training the dynamic router; and 3) fine-tuning the sparsity scheduler. First, we search for the sparsity scheduler (the pruning ratio for each block) based on a customized static router. Next, we fix the sparsity scheduler and fine-tune the dynamic router using our proposed distillation, guided loss, and sparsity loss methods. Subsequently, we fix the router and re-search the pruning scheduler similar to the first step. Note that these steps can be repeated $N$ times to further enhance performance; however, in our approach, we only repeat them once for simplicity

of training. Finally, our dynamic router can adaptively provide effective pruning strategies for each block of the model.

**Sparsity Scheduler Searching with Static Router.** Given the overall sparsity, we first search for an initial sparsity scheduler using our proposed customized static router. Inspired by the work (Xiao et al., 2023) that indicates that the initial token plays a key role in the window attention of LLMs for long-text inference, and in this work, we have found that the first and the last few tokens in a sequence are typically important for model performance. Therefore, we construct a static router that ranks the importance of tokens $x$ at the $i$-th block as $\{x_0^i, x_{L-1}^i, x_{L-2}^i, ..., x_1^i\}$. This means that the first token and the last few tokens will be prioritized for computation. In the pruning process, the top-$k$ unimportant tokens will be skipped to meet the sparsity requirement and passed directly to the next block. Next, we utilize Genetic Algorithm (Harada & Alba, 2020) to search for the optimal sparsity scheduler for each block of the large language models (LLMs). The objective of this search strategy is to allocate the sparsity ratio for each block while maintaining overall model sparsity and maximizing the evaluation accuracy. We formulate this search objective as Equation 3.

$$S^* = \arg\max_S \text{Accuracy}(\theta(\mathcal{R}(S), X), Y) \quad \text{s.t.} \quad \sum s_i = P \tag{3}$$

where the $\theta(\mathcal{R}(S), X)$ indicates that the LLM model $\theta$ works with a router $\mathcal{R}$ assigned a sparsity ratio configuration $S$ and is fed with input $X$ for prediction.

Our method decouples the sparsity allocation and router tuning processes. The initial search using the static router provides a good preliminary pruning configuration, which serves as the starting point for training the router. This design is simple yet effective, surpassing existing state-of-the-art (SOTA) methods, as shown in Table 1 (FTP (static)). More results of FTP (static) are in Appendix A.4.

**Dynamic Router.** We construct a lightweight dynamic router to control whether the tokens in each block need to be computed or skipped during the forward process. Recent router-based methods (Raposo et al., 2024) leverage hidden states to predict pruning configurations. However, we argue that this approach is not effective, as hidden states are high-dimensional abstract features that require heavy network fitting, leading to increased training and inference costs, and potentially degrading generalization. Thus, it is not suitable for LLM pruning tasks. To address this issue, we propose four low-dimensional factors that are weakly correlated with token hidden states but are related to token redundancy.

Specifically, for a set of token embeddings in a sequence of length $L$ for the $i$-th block, our proposed factors for the router can be represented as:

$$H^i = \{\boldsymbol{h}_j^i \mid j \in \mathbb{N}, 1 \le j \le L\} = \{(p^j, s_a^j, r_a^j, s_r^i) \mid j \in \mathbb{N}, 1 \le j \le L\}, \tag{4}$$

where $\boldsymbol{h}_j^i$ is the hybrid input vector of the $j$-th token in the sequence, which is a 4-dimensional vector that includes the token position $p^j$, absolute attention scores $s_a^j$, relative attention score rank $r_a^j$, and sparsity requirements $s_r^i$ of the $i$-th block.

Previous work (Xiao et al., 2023) has revealed that tokens at different positions in a sequence are of varying importance. Thus, we incorporate the token position to decide whether to prune a token. Additionally, attention scores represent the degree of association between a token and other tokens, making it a crucial pruning factor. If a token is highly associated with others, it can be replaced, indicating its redundancy. Moreover, we introduce relative attention score rank to measure the relative importance of tokens, along with sparsity requirements to control the pruning rate. This enables our dynamic router to allocate pruning configurations from a global perspective. Building on our effective factors, we design a lightweight router consisting of a two-layer MLP that takes a 4-dimensional factor vector as input and produces a 2-dimensional output importance score. The output importance score $\boldsymbol{o}_j^i$ is normalized by a softmax operation and represents the probability of computing the $j$-th input token of the $i$-th block in the forward process. This score is processed by the $\arg\max$ operation and discretize it into a gate $\boldsymbol{g} \in \{0, 1\}$. This gate is used to control whether to skip ($\boldsymbol{g} = 0$) or compute ($\boldsymbol{g} = 1$) the token in the block. However, the $\arg\max$ operation is non-differentiable. Therefore, we utilize the Straight-Through (ST) Estimator Jang et al. (2016) during the training phase to approximate the real gradient $\nabla_\theta \boldsymbol{g}$ with the gradient of the soft prediction $\nabla_\theta \boldsymbol{s}^i$. During training or inference, the proposed inputs of all tokens from a block are fed into the router to obtain the predicted importance scores for all the tokens. Note that all blocks share the same router, enhancing the router's generalization ability.

**Training Losses.** We propose three training losses to enhance pruning performance: guide loss, sparsity constraint loss, and knowledge distillation loss. Specifically, to accelerate the training process of the learnable router, we introduce guide loss as a warm-up constraint at the beginning of training. The guide loss leverages the static router constructed in Step 1 as a teacher model to guide the student model (i.e., the dynamic router) in producing reasonable importance score predictions during the early stages of training. This is achieved using a binary cross-entropy (BCE) loss. The sparsity constraint loss is employed to align the predicted sparsity with the required sparsity of the blocks. The predicted sparsity ratio for each block is obtained via the summation of skipping tokens based on the gate $\boldsymbol{g}$. The constraint loss imposes a penalty on the router only if the predicted sparsity ratio is less than the assigned sparsity ratio as follows:

$$\mathcal{L}_s = \sum_{i}^{N}(\max(s_r^i - \frac{1}{L}\sum_{j}^{L}(1 - \boldsymbol{g}_j^i), 0)) \tag{5}$$

where $N$ is the number of the LLM's blocks, $\boldsymbol{g}^i$ is the predicted discrete state of the token sequence in the $i$-th block of the LLM and $s_r^i$ is the required sparsity ratio of that block.

Moreover, the knowledge distillation loss is utilized to improve the accuracy of the pruned model by aligning the predictions between the original and pruned models using mean squared error (MSE) loss. We apply the distillation loss only at the output of the last block in the LLM for all tokens. These losses are combined to optimize the learnable router with different loss weights, resulting in the final loss as follows:

$$\mathcal{L}(X, Y; \theta, \mathcal{R}) = \lambda_d \mathcal{L}_d + \lambda_s \mathcal{L}_s + \lambda_g \mathcal{L}_g \tag{6}$$

where $\lambda_d$, $\lambda_s$ and $\lambda_g$ are the loss weights of distillation loss $\mathcal{L}_d$, sparsity constraint loss $\mathcal{L}_s$ and guide loss $\mathcal{L}_g$, respectively. $\theta$ and $\mathcal{R}$ denote the parameters of the LLM and dynamic router, respectively. The loss weight $L_g$ is initially set to 1 and gradually decays to 0 as the training progresses to halfway through the total number of iterations.

**Training and Inference.** In the training phase, we first utilize the sparsity scheduler search strategy to obtain an effective initial sparsity ratio configuration based on our proposed static router. During the dynamic router training stage, we maintain an attention scores table to record the latest attention scores of all tokens and update the scores of the computed tokens in sequences. Once the router is well-trained, we fine-tune the initial sparsity ratios using our sparsity ratio optimization strategy based on the learnable router. Subsequently, the learnable router, coupled with the refined sparsity ratios, can be used to accelerate the LLM.

## 4 EXPERIMENTS

### 4.1 EXPERIMENTAL SETTINGS

**Models and Baselines.** We apply FTP to LLaMA2-7B, LLaMA2-13B (Touvron et al., 2023), LLaMA3-8B (Dubey et al., 2024), and Qwen1.5-7B (Bai et al., 2023), with initialization by non-instruct-tuning pretrained weights. To assess the effectiveness of our approach, we benchmark it against state-of-the-art structured pruning techniques, including LLMPruner (Ma et al., 2023), SliceGPT (Ashkboos et al., 2024), LaCo (Yang et al., 2024), ShortGPT (Men et al., 2024), Relative Magnitude(RM) (Samragh et al., 2023), and BlockPruner (Zhong et al., 2024). LLMPruner and SliceGPT primarily target pruning through reductions in embedding dimensions, whereas LaCo, ShortGPT, RM, and BlockPruner focus on depth pruning strategies.

**Datasets.** Following previous works, we use the Alpaca (Taori et al., 2023) for training, and validation on these well-known benchmarks: HellaSwag (Zellers et al., 2019), MMLU (Hendrycks et al., 2020), ARC-easy, ARC-challenge (Clark et al., 2018), WinoGrande (Sakaguchi et al., 2021); specifically, we utilize the WinoGrande to optimize the sparsity ratios due to its various token length. We report the accuracies together with average accuracy retention percentages on these benchmarks.

**Implement Details.** We train 10,000 and 50,000 steps for 7/8B and 13B models, respectively, with a batch size of 1. We utilize the AdamW optimizer with a learning rate of 1e-4. All experiments are conducted on a single AMD MI250 GPU with 64GB of memory, taking approximately 1 hour for the router training phase. We provide more details about implementation in Appendix A.3.

Table 1: **Downstream tasks performance.** FTP surpasses all the competitors under comparable sparsity constraints. MMLU uses a 5-shot evaluation, and other tasks are all 0-shot.

| Model | Method | Ratio (%) | ARC-c | ARC-e | HellaSwag | MMLU | WinoGrande | Avg. Percentage (%) |
|---|---|---|---|---|---|---|---|---|
| LLaMA2-7B | Dense | 0 | 46.16 | 74.54 | 75.99 | 45.39 | 69.06 | 100 |
| | LaCo | 21.02 | 35.84 | 55.39 | 54.08 | - | 60.46 | 77.67 |
| | RM | 21.02 | 22.53 | 34.43 | 29.22 | - | 49.25 | 51.19 |
| | LLMPruner | 27.0 | - | - | 60.21 | 23.33 | - | 65.32 |
| | SliceGPT | 21.45 | 37.12 | 63.64 | 56.04 | - | 59.91 | 81.57 |
| | ShortGPT | 27.0 | 32.68 | 48.61 | 56.15 | 44.51 | 64.33 | 80.22 |
| | BlockPruner | 20.99 | 35.92 | 61.20 | 66.04 | - | 64.09 | 84.91 |
| | **FTP (static)** | 22.0 | 44.88 | 72.31 | 72.66 | 45.83 | **69.53** | 98.30 |
| | **FTP** | 22.0 | **45.31** | **73.06** | **74.46** | **46.15** | 69.22 | **99.21** |
| | **FTP** | 30.0 | 43.65 | 72.31 | 67.37 | 46.07 | 68.97 | 96.32 |
| LLaMA2-13B | Dense | 0 | 49.23 | 77.36 | 79.36 | 54.94 | 72.14 | 100 |
| | LaCo | 24.37 | 34.56 | 54.34 | 60.44 | - | 59.27 | 74.69 |
| | RM | 24.37 | 41.98 | 66.12 | 66.80 | - | 66.61 | 86.81 |
| | SliceGPT | 21.52 | 42.41 | 68.52 | 60.71 | - | 65.59 | 85.53 |
| | ShortGPT | 24.60 | 42.92 | 63.55 | 69.27 | 53.83 | 69.85 | 90.28 |
| | BlockPruner | 24.31 | 40.53 | 63.55 | 71.93 | - | 70.40 | 88.18 |
| | **FTP (static)** | 25.0 | 47.95 | 74.58 | 76.65 | 54.51 | 71.19 | 97.66 |
| | **FTP** | 25.0 | **48.98** | **75.55** | **77.49** | **54.56** | **72.22** | **98.84** |
| | **FTP** | 30.0 | 48.38 | 74.75 | 75.99 | 54.47 | 71.67 | 97.83 |
| Qwen1.5-7B | Dense | 0 | 42.66 | 62.16 | 76.92 | 60.52 | 66.46 | 100 |
| | LaCo | 20.97 | 32.85 | 46.89 | 56.35 | - | 58.64 | 78.48 |
| | RM | 20.97 | 28.58 | 54.17 | 42.00 | - | 49.88 | 70.95 |
| | ShortGPT | 21.88 | 33.79 | 48.44 | 63.09 | 49.54 | 60.93 | 82.54 |
| | BlockPruner | 21.83 | 33.02 | 53.49 | 57.29 | - | 56.99 | 80.92 |
| | **FTP (static)** | 22.0 | 43.52 | 62.71 | 71.89 | 60.26 | 65.19 | 98.80 |
| | **FTP** | 22.0 | **43.69** | **62.81** | **74.02** | **60.86** | **67.32** | **100.03** |
| | **FTP** | 30.0 | 40.96 | 59.60 | 68.47 | 60.77 | 65.67 | 96.03 |

## 4.2 MAIN RESULTS

**Compare with SOTA Methods.** As shown in Table 1, FTP variants demonstrate superior performance across five public benchmarks: ARC-c, ARC-e, HellaSwag, MMLU, and WinoGrande, covering various tasks such as reasoning, language understanding, knowledge retention, and examination capacity. Our FTP method consistently outperforms other SOTA pruning methods, such as BlockPruner and ShortGPT, across models like LLaMA2-7B, LLaMA2-13B, and Qwen1.5-7B, demonstrating notable improvements in average performance. For example, at a 22% sparsity ratio, FTP achieves 99.21% on LLaMA2-7B, compared to BlockPruner's 84.91%. Similarly, ShortGPT only reaches 80.22% on LLaMA2-7B at a 27% sparsity ratio, while FTP attains 96.32% at an even higher 30% sparsity ratio. These results highlight FTP's remarkable ability to effectively prune tokens across different blocks in large language models while maintaining high accuracy across various tasks.

Furthermore, Table 1 demonstrates our proposed static router outperforms other SOTA methods, owing to the combined effectiveness of the sparsity scheduler and the efficient routing strategy implemented within each block. Moreover, our dynamic router surpasses even this performance, which can be attributed to the inherent advantages of dynamic routing, the innovative trainable router, and the specifically designed input and optimization losses.

Table 2: **Various sparsity ratios.** FTP still maintains relatively roubst performance at higher sparsity ratio (40%), and is even better than BlockPruner, ShortGPT and other methods on LLaMA2-7B with a sparsity ratio of 22% (in Table 1).

| Model | Ratio (%) | ARC-c | ARC-e | HellaSwag | MMLU | WinoGrande | Avg. Percentage (%) |
|---|---|---|---|---|---|---|---|
| LLaMA2-7B | 0 | 46.16 | 74.54 | 75.99 | 45.39 | 69.06 | 100 |
| | 30 | 43.65 | 72.31 | 67.37 | 46.07 | 68.97 | 96.32 |
| | 40 | 40.02 | 70.01 | 62.67 | 46.56 | 66.03 | 92.26 |
| LLaMA2-13B | 0 | 49.23 | 77.36 | 79.36 | 54.94 | 72.14 | 100 |
| | 30 | 48.38 | 74.75 | 75.99 | 54.47 | 71.67 | 97.83 |
| | 40 | 45.22 | 70.88 | 66.50 | 54.57 | 70.40 | 92.84 |
| LLaMA3-8B | 0 | 53.33 | 77.69 | 79.19 | 65.28 | 72.85 | 100 |
| | 30 | 48.63 | 73.36 | 62.41 | 64.29 | 69.69 | 91.71 |
| | 40 | 43.00 | 67.17 | 54.89 | 63.72 | 69.30 | 85.83 |
| Qwen1.5-7B | 0 | 42.66 | 62.16 | 76.92 | 60.52 | 66.46 | 100 |
| | 30 | 40.96 | 59.60 | 68.47 | 60.77 | 65.67 | 96.03 |
| | 40 | 36.15 | 53.03 | 62.59 | 60.83 | 59.04 | 88.15 |

Table 3: Overall comparisons of sparsity allocations on LLaMA2-7B with 30% sparsity.

| Method | ARC-c | MMLU | Avg. Percentage |
|---|---|---|---|
| Uniform | 26.02 | 40.50 | 72.80 |
| BI score based | 34.81 | 45.29 | 87.60 |
| SS w.o finetune | 40.96 | 45.67 | 94.68 |
| SS w.finetune | **43.65** | **46.07** | **98.03** |

Table 4: Overall comparisons of different routers on LLaMA2-7B with 30% sparsity.

| Method | ARC-c | MMLU | Avg. Percentage |
|---|---|---|---|
| Recurrent router | 40.23 | 45.53 | 93.73 |
| Local router | 38.21 | 43.69 | 89.52 |
| Global router | **43.65** | **46.07** | **98.03** |

Notably, even at a higher sparsity ratio (30%), FTP surpasses other methods. On LLaMA2-13B, FTP achieves an average accuracy of 97.83% at 30% sparsity, significantly outperforming Block-Pruner (88.18%) and ShortGPT (90.28%). This underscores FTP's robustness in maintaining model performance despite a substantial reduction in the number of tokens processed per block. Moreover, at a 22% sparsity ratio on Qwen1.5-7B, FTP's pruning results even exceed those of the dense model across the five benchmarks, further showcasing its efficiency.

**Higher Sparsity on Different Models.** In Table 2, we examine the impact of increasing sparsity on FTP's performance. At a 40% sparsity ratio, FTP maintains an impressive performance range of 85% to 93% across various models and benchmarks. Specifically, on LLaMA2-7B, FTP achieves 92.26% at 40% sparsity, significantly outperforming BlockPruner (84.91%) at 22% sparsity and ShortGPT (80.22%) at 27% sparsity. This indicates that FTP not only manages higher sparsity more effectively but also surpasses other methods even under more conservative pruning settings. The analysis of performance degradation shows that even when reducing the number of tokens by 40%, FTP's performance still remains strong compared to other SOTA methods. The comparison with other methods such as BlockPruner and ShortGPT is particularly telling. Additionally, FTP demonstrates consistent high performance across different model sizes, as seen when comparing LLaMA2-7B with LLaMA2-13B. In Table 2, FTP achieves an average performance of 96.32% at 30% sparsity ratio on LLaMA2-7B, and a comparable 97.87% at 30% sparsity ratio on LLaMA2-13B. This indicates that FTP is robust in handling sparsity across models of varying sizes, and scales effectively without significant performance degradation. Such consistency across models indicates that FTP is highly scalable and reliable for deployment in larger models where computational efficiency is critical.

### 4.3 ABLATION STUDY

**Effect of the Sparsity Scheduler.** In Section 3.1, we highlight the varying sensitivity of blocks at different depths to token pruning and introduce a GA-based sparsity scheduler (SS) to determine the optimal sparsity ratios for all blocks in the LLM, while meeting the overall pruning requirement. Table 3 demonstrates the effectiveness of our sparsity allocation compared to other strategies. Notably, a uniform (average) sparsity distribution results in a 24.25% performance drop compared to our approach. Even sparsity allocation method based on weight initialization, such as the BI score (Men et al., 2024), shows a performance gap of about 10% when compared to our optimized sparsity allocation. Additionally, we further enhance the sparsity allocation by post-tuning, after the trainable router has been well-trained using the initial allocation.

**Effect of the Designed Input.** The core idea of our method is to rank tokens based on their predicted importance and skip the less significant ones within a block. The input design for the router plays a crucial role in determining the outcome. In this section, we compare various inputs in Table 5 to illustrate the effectiveness of our designed input. Previous work (Raposo et al., 2024) uses hidden states from each block as the sole feature for the router's decision-making. Thus, we directly compare the hidden states as input with our designed input. As shown in Table 5, our designed input significantly outperforms both the hidden states and combinations that include hidden states. Additionally, we conduct ablation studies to assess the individual elements of the designed input, confirming the importance of all components.

**Effect of the Proposed Router.** We also explore different structural designs for the token router. As shown in Table 4, we compare a recurrent router and a local router with our global router. The recurrent router uses an LSTM model, treating each block as a step and predicting token routing decisions based on the designed input, along with the previous block's decision and token importance. Its performance is lower than that of the global router, likely due to the accumulation of incorrect judgments and importance estimates from previous blocks. The local router, which shares the same

Table 5: Overall comparisons of different inputs on LLaMA2-7B with 30% sparsity. DI indicates our designed input.

| Method | ARC-c | MMLU | Avg. Percentage |
|---|---|---|---|
| Dense | 46.16 | 45.39 | 100 |
| Hidden states | 33.87 | 44.78 | 86.02 |
| DI w. hidden states | 34.57 | 44.99 | 87.01 |
| DI w.o. position | 30.72 | 44.59 | 82.39 |
| DI w.o. attntion score | 41.21 | 45.43 | 94.68 |
| DI w.o. attntion rank | 42.13 | 45.15 | 95.37 |
| DI w.o. sparsity | 38.51 | 45.79 | 92.15 |
| DI | **43.65** | **46.07** | **98.03** |

Table 6: Inference speedup of our FTP in LLaMA2-7B on different settings including different sparsity ratios and token lengths.

| Method | Ratio(%) | Token Length | Infer Speedup |
|---|---|---|---|
| Dense | 0 | 1000 | 1.0 × |
| FTP | 30 | 1000 | 1.28 × |
| FTP | 40 | 1000 | 1.41 × |
| Dense | 0 | 2000 | 1.0 × |
| FTP | 30 | 2000 | 1.39 × |
| FTP | 40 | 2000 | 1.61 × |

structure as the global router but assigns an independent router to each block, also underperforms. This may be because the global router offers a more comprehensive view of block interdependencies, whereas the local router focuses primarily on optimizing each block individually.

## 4.4 MORE ANALYSIS

**Inference Speedup.** The forward computation of transformer blocks constitutes a large portion of the inference time, whereas the computational cost of our global router—comprised of just a two-layer MLP with a 4-channel input—accounts for only a small fraction of the overall inference time. The computational advantage of our approach grows as the sequence length increases. When the router selects specific tokens to skip within a block, the length of the token sequence involved in attention computation is reduced, thereby decreasing computational complexity at a quadratic rate. Additionally, the feed-forward network (FFN) costs are eliminated for the skipped tokens. We evaluate the speed performance under different configurations using the Alpaca dataset as input prompts on LLaMA2-7B. By adjusting the token length and calculating the average inference time, we compare the speedups. As shown in Table 6, FTP achieves a higher acceleration ratio with longer token sequences, even at the same sparsity level. With the increasing importance of ultra-long context technology in the development of large language models (LLMs), FTP gains a significant advantage as sequence lengths grow.

**Compatible with Key and Value (KV) Cache.** The KV cache stores key and value representations for each token across different transformer blocks, enabling faster retrieval and computation during autoregressive inference, especially in tasks like text generation. This approach reduces redundant computation by reusing key-value pairs from previous tokens, making inference more efficient. As a result, the primary computational cost shifts to focus mainly on the last token in the sequence, which includes feed-forward network (FFN) operations and attention computations with other tokens in the sequence. Since our method does not impose a sparsity constraint on the last token in the depth dimension, but rather on the entire token sequence, the KV cache reduces the acceleration benefits gained from token-wise pruning. This is because our method prioritizes the forward computation of the last token. To address this, we modify the router to impose constraints on the sparsity ratio of the last token in the depth dimension. Specifically, we introduce a threshold (0.5), determined through evaluations on the WinoGrande dataset, that governs the sparsity of the last token. If the router's predicted score for the last token exceeds the threshold, it performs computation within the block; otherwise, it is skipped. As demonstrated in Appendix A.8, the pruning results show virtually no performance loss.

## 5 CONCLUSION

In this paper, we present a fine-grained token-wise pruning framework for the LLMs, which can outperform other SOTA LLM pruning methods without the retraining process. Our proposed token-wise pruning framework is structured around three key steps: first, we conduct an initial sparsity search utilizing a static router to determine the appropriate sparsity allocation. Next, we train a dynamic router informed by our four proposed factors and three distinct loss functions. Finally, we fine-tune the sparsity scheduler using the trained router. Comprehensive experiments underscore the importance of each component in improving the overall effectiveness of our approach. The results reveal that our method significantly outperforms other SOTA methods, further demonstrating its superiority.

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

# A  APPENDIX

## A.1  ETHICS STATEMENT

This research focuses on improving the efficiency of large language models (LLMs) through fine-grained token-wise pruning, with the goal of reducing computational costs during inference while maintaining model performance. Our work does not involve human subjects or the collection of sensitive data, and thus, does not raise concerns related to privacy, security, or legal compliance. In terms of dataset usage, we primarily evaluate our approach using publicly available benchmark datasets, such as WinoGrande, ARC-c, and MMLU, which are widely used in the field. We ensure compliance with the licensing and usage terms of these datasets. No personally identifiable information or sensitive data is included in our experiments. We are mindful of the potential societal impact of our research, especially concerning the deployment of LLMs in real-world applications. While the techniques proposed in this work can lead to more efficient LLM deployments, which may lower computational resource requirements and costs, we recognize that LLMs, in general, can perpetuate biases present in their training data. Our research focuses on improving efficiency and does not directly address fairness or bias in language models. However, we acknowledge the importance of addressing these issues in future work. Additionally, all authors have no conflicts of interest influencing the research presented in this paper.

## A.2  REPRODUCIBILITY STATEMENT

Comprehensive descriptions of the datasets used in our experiments are provided, please refer to the Section 4.1. We report the software version and hardware environments, and related hyper-parameters in training and validation, please refer to the Section A.3 and 4.1. In Section 3.1, we introduce the LLM architecture and discuss the token redundancy in Section 3.1. The implementation details of the sparsity scheduler are provided in Section 3.2, followed by a description of the static router in Section 3.2 and the dynamic router in Section 3.2. Finally, the loss formulations are presented in Section 3.2. We report the ablation study results in Section 4.3. We believe these efforts will facilitate the replication and verification of our findings by other researchers. The research is conducted with full adherence to research integrity standards, and all relevant documentation, code, and experimental results will be made available after obtaining a public license.

## A.3  IMPLEMENTATION DETAILS

The token router consists of a two-layer MLP, with a hidden size of 64 and an output size of 2. In our experiments, the hyperparameters of the loss function (i.e., $\lambda_d$, $\lambda_s$, and $\lambda_g$) are all initially set to 1. During the sparsity optimization stage, we use a population of 50 sparsity configurations, with 10 generations and a mutation probability of 0.2. The sparsity optimization process takes approximately 2 hours. The training and sparsity optimization processes are implemented using ROCm 6.1, Torch 2.3, and Torchtune 2.0. We employ lm-eval to evaluate the benchmarks. For consistency and fairness, FP32 precision is uniformly used during both training and testing. All benchmarks are evaluated using the "Acc norm" score by default, and the average percentage reflects the average score across all benchmarks (i.e., pruned/dense model performance).

## A.4  MORE ANALYSIS ON STATIC FTP

**Compared to the random selection.** After obtaining the sparsity configuration under the overall sparsity ratio of 30%, we compare the performance between random token selection and static FTP, as shown in Table 7. In the random selection, we randomly choose the same number of tokens as the static FTP for skipping, while keeping within its sparsity ratio limits. The random selection is cross-validated 5 times, with results averaged across trials. Notably, static FTP consistently outperforms random selection, highlighting the critical importance of token position in selection. Despite this, random selection achieves nearly 70% performance, due to the underlying block configuration derived from the search process. This underscores the significance of the sparsity scheduler in maintaining performance.

**Priority Token Retained.** We conduct a further investigation into the token importance. A comparison between the random token selection approaches in rows 3 and 4 reveals that performance

improves when the first token is retained. This is further supported by the results in row 5 of Table 7, where the firstly retaining of the second token leads to a performance drop in the static FTP approach. These findings highlight the critical importance of the first token selection. Furthermore, we observe a significant drop in performance when introducing random perturbations into the final static decision process. Specifically, we randomly select 10% of tokens from the sequence (take 5% from skip tokens, and 5% from updated tokens) and swap their decision flags to maintain the block sparsity ratio. This highlights the sensitivity of token selection within the model. Nevertheless, our dynamic FTP outperforms the static version, demonstrating the robustness and efficacy of the dynamic routing mechanism.

Table 7: Comparisons of different static routers with 30% sparsity.

| Method | Priority Retained Token ID | ARC-c | MMLU | Avg. Percentage |
|---|---|---|---|---|
| Dense | - | 46.16 | 45.39 | 100 |
| Random selection | - | 28.92 | 34.16 | 68.96 |
| Random selection | 1st | 32.17 | 34.84 | 73.22 |
| FTP (static)* | 2nd | 31.91 | 34.54 | 72.61 |
| FTP (static) w. perturbation | 1st | 40.19 | 43.95 | 91.95 |
| FTP (static) | 1st | 43.26 | 46.09 | 97.63 |

## A.5 ATTENTION SCORE

Define the $Q \in \mathbb{R}^{L \times d \times N}$ and $K \in \mathbb{R}^{L \times d \times N}$, where the $L$ is the sequence lengths of the query and key in attention. The $N$ is the head number of the multi-head attention. The attention score $A_s \in \mathbb{R}^L$ can be formulated as following:

$$A = \frac{QK^T}{\sqrt{d}}$$

$$A_s = \frac{1}{L} \sum_{j=1}^{L} A_{i,j}, i = 1, 2, \ldots, L \tag{7}$$

After obtaining the $A \in \mathbb{R}^{L \times L \times N}$, we execute a mean operation in head dimension $N$, then we obtain the $A_s$ by a mean operation in the dimension of the key length. The attention score can reflect the relationships among the tokens, which is an important factor as input for the learnable router.

## A.6 PSEUDO CODE OF GA-BASED SPARSITY SCHEDULER

We introduce the details of the GA-based sparsity scheduler via pseudo-code in Algorithm 1. The GA-based approach aims to find an optimal block-wise sparsity configuration, $S^*$, for an LLM $\mathcal{M}$, that satisfies a target overall sparsity ratio $P_{\text{overall}}$, while maximizing model performance. The process begins by generating an initial population $\mathcal{P}$ of candidate configurations, where each configuration $S_i$ is sampled from the search space $\mathcal{S}_{\text{space}}$, ensuring $\sum s_i = P_{\text{overall}}$. Each configuration is assessed by applying it to the LLM and measuring the model's accuracy on the evaluation dataset, $\mathcal{D}_{\text{eval}}$. Following these evaluations, the configurations are ranked by accuracy, with the highest-performing ones selected for reproduction.

In each iteration, parents are selected to produce offspring through crossover and mutation. Mutation is applied with a probability of $p_{\text{mutate}}$ to introduce diversity while preserving the overall sparsity constraint. The offspring are evaluated and replaced with the worst-performing configurations in the population. This process is repeated for $T_{\text{max\_iter}}$ iterations, with the population progressively evolving towards an optimal solution. The final configuration, $S^*$, which achieves the highest accuracy, is returned as the optimal sparsity configuration, effectively balancing model performance and computational efficiency.

## A.7 SPARSITY RATIO RESULTS

As shown in Table 8, we report the block-wise sparsity ratio details obtained from the scheduler. Note that, the block ID is started from 0. We join the (block number - 2) blocks into the scheduler,

---

**Algorithm 1** GA-Based Sparsity Scheduler

---

**Input:**
  $\mathcal{M}$: Pretrained LLM
  $P_{\text{overall}}$: Target sparsity ratio
  $\mathcal{D}_{\text{eval}}$: Evaluation dataset
  $\mathcal{S}_{\text{space}}$: Search space of block-wise sparsity
  $\mathcal{T}_{\text{max\_iter}}$: Max iterations for GA
**Output:**
  $S^*$: Optimal block-wise sparsity ratio configuration

1: Initialize population $\mathcal{P}$ of block-wise sparsity configurations $\{S_i\}$ from $\mathcal{S}_{\text{space}}$, where $\sum s_i = P_{\text{overall}}$.
2: Evaluate each $S_i$ in $\mathcal{P}$ by applying it to $\mathcal{M}$ on $\mathcal{D}_{\text{eval}}$ and record Accuracy$(\mathcal{M}_{S_i}, \mathcal{D}_{\text{eval}})$.
3: Sort $\mathcal{P}$ by accuracy and select top configurations.
4: Set $t = 0$.
5: **while** $t < \mathcal{T}_{\text{max\_iter}}$ **do**
6:     Select parents from $\mathcal{P}$ based on performance.
7:     Crossover selected parents to generate new configurations.
8:     Mutate offspring configurations with probability $p_{\text{mutate}}$, ensuring $\sum s_i = P_{\text{overall}}$.
9:     Evaluate offspring by computing Accuracy$(\mathcal{M}_{S_{\text{offspring}}}, \mathcal{D}_{\text{eval}})$.
10:    Replace worst-performing configurations with the best offspring.
11:    Sort updated $\mathcal{P}$ by accuracy.
12:    $t \leftarrow t + 1$
13: **end while**
14: **return** $S^*$ with the highest accuracy from the final population.

---

e.g., 32 blocks in Llama2-7B and 30 blocks involve optimization. Note that, the sparsity ratio of blocks not mentioned in this table are default 0.

Table 8: Block-wise sparsity ratios obtained by sparsity scheduler for overall 30% sparsity.

| Model | Results (Block ID: Sparsity ratio) |
|---|---|
| LLama-2-7B (Initial) | 16: 0.2596, 17: 0.3987, 18: 0.4808, 19: 0.5481, 20: 0.5451, 21: 0.6642, 22: 0.682, 23: 0.7337, 24: 0.7589, 25: 0.7973, 26: 0.7766, 27: 0.7996, 28: 0.7862, 29: 0.7729, 30: 0.5962 |
| LLama-2-7B (Finetuned) | 13: 0.1708, 14: 0.1904, 15: 0.1912, 16: 0.1839, 17: 0.3372, 18: 0.4277, 19: 0.5019, 20: 0.4986, 21: 0.6299, 22: 0.6494, 23: 0.7065, 24: 0.7342, 25: 0.7766, 26: 0.7538, 27: 0.7791, 28: 0.7644, 29: 0.7497, 30: 0.5548 |
| LLama2-13B (Initial) | 11: 0.0693, 12: 0.1014, 13: 0.1182, 14: 0.1477, 15: 0.1595, 16: 0.1164, 17: 0.1401, 18: 0.1283, 19: 0.2169, 20: 0.2363, 21: 0.3318, 22: 0.3019, 23: 0.4579, 24: 0.7259, 25: 0.659, 26: 0.5836, 27: 0.5282, 28: 0.5267, 29: 0.6702, 30: 0.5661, 31: 0.658, 32: 0.6851, 33: 0.6721, 34: 0.6825, 35: 0.6053, 36: 0.8427, 37: 0.6111, 38: 0.5934 |
| LLama2-13B (Finetuned) | 12: 0.0171, 13: 0.0148, 14: 0.0476, 15: 0.0795, 16: 0.1144, 17: 0.1467, 18: 0.2193, 19: 0.4022, 20: 0.4383, 21: 0.4738, 22: 0.5108, 23: 0.6531, 24: 0.5355, 25: 0.597, 26: 0.5807, 27: 0.599, 28: 0.627, 29: 0.6175, 30: 0.6068, 31: 0.6059, 32: 0.6012, 33: 0.6019, 34: 0.613, 35: 0.6007, 36: 0.6127, 37: 0.6111, 38: 0.4786 |

## A.8 THE RESULTS OF SUPPORTING KV CACHE

As depicted in Section 4.4, we introduce a specific threshold to constrain the sparsity ratio for the last token in the depth dimension of LLMs. Apart from the last token, the router's decisions for the other tokens continue to follow the original approach, selecting the required ratio of remaining tokens to be skipped based on the predicted score within each block. Thus, our method, incorporating KV cache modifications, enforces two sparsity constraints: token sparsity across the sequence and last token sparsity in the depth dimension. However, the threshold strategy can not strictly constrain the sparsity of the last token in different input sequences.

Furthermore, we introduce a strict sparsity constraint strategy, combined with the threshold strategy during autoregressive decoding, to consistently ensure that the sparsity target for the last token is achieved. This method monitors the sparsity of the last token across the depth dimension and halts the processing of additional blocks once the target sparsity is reached. If the cumulative sparsity reaches the target before finishing all block computations, subsequent blocks for the last token are

required to undergo forward computation. Meantime, it also monitors the number of the remaining blocks waiting for computation together with the current sparsity of the last token to ensure the final sparsity can meet the target sparsity. If the combination of the ratio of remaining blocks and the current sparsity is close to the target, the subsequent blocks should be skipped to guarantee that the final sparsity meets the intended goal.

Table 9: Performance comparisons of different methods on LLaMA2-7B.

| Method | Ratio (%) | ARC-c | MMLU | Avg. Percentage | PPL |
|---|---|---|---|---|---|
| Dense | 0 | 46.16 | 45.39 | 100 | 5.47 |
| ShortGPT | 21.02 | 36.09 | 44.51 | 88.04 | 18.45 |
| BlockPruner | 21.99 | 37.29 | - | 80.78 | 11.51 |
| FTP | 22.0 | 45.31 | 46.15 | 99.90 | 11.14 |
| FTP (threshold) | 21.92 | 45.52 | 46.35 | 100.35 | 11.12 |
| FTP (strict constraint) | 22.10 | 45.30 | 46.12 | 99.86 | 11.14 |

As shown in Table 9, the pruning results, along with the supporting KV cache modifications, demonstrate virtually no performance loss compared to the original results on the ARC-c and MMLU benchmarks. Moreover, the perplexity (PPL) results further demonstrate the robustness of our method in text generation, with a PPL of 11.12 using a threshold strategy to support KV cache, which surpasses the other SOTA methods, indicating that text generation performance remains stable. Additionally, applying the strict sparsity constraint ensures that the overall sparsity target can be met, with a PPL of 11.14 and minimal accuracy impact, confirming that our method is effectively compatible with the KV cache.

