# OpenReview forum: "FTP: A Fine-grained Token-wise Pruner for Large Language Models via Token Routing"
_ICLR.cc/2025/Conference — ICLR 2025 Conference Withdrawn Submission_

### Official Review · Reviewer_rYAa · 2024-11-02

**Soundness:** 2
**Presentation:** 2
**Contribution:** 2
**Rating:** 3
**Confidence:** 5

**Summary:**

This study proposes a fine-grained token-wise pruning approach for the LLMs, which presents a learnable router to adaptively identify the less essential tokens and skip them across model blocks to reduce computational cost during inference. Comparative experiments were conducted on different benchmarks, performing better than several existing works.

**Strengths:**

1. This paper is well-organized and easy to follow.
2. Comparative experiments were conducted on different benchmarks, performing better than several existing works.

**Weaknesses:**

1. The scientific issues of the proposed paper are unclear. Existing prune methods of LLMs aim to remove the redundant module. This paper fails to discuss the shortcomings or deficiencies of existing work and cannot present the paper's main contribution.
2. The novelty of this paper is limited. Several works have discussed the Redundant issue, e.g., [1], [2], which cannot be considered one of the paper's main contributions.
3. Pruning aims to remove unimportant parts, not redundant parts. High redundancy comes from high similarity, which is highly related to calculating multi-head attention (attention scoring comes from similarity). Obviously, removing these tokens affects the subsequent multi-head attention and FFN calculations.
4. The authors spent much space describing the multi-head attention mechanism in Section 3.1, which has nothing to do with the author's tokenwise prune. All symbols in Eq. (1) and (2) no longer appear or are even mentioned in the following content. Since the page is limited, presenting more in-depth experimental content is more meaningful.
5. Equal balance co-efficiencies in Eq. (6) result in pruning losses exceeding task training losses, leading the model to tend towards pruning rather than achieving good inference performance.
6. The experiment is too routine to demonstrate the effectiveness and rationale of the proposed dynamic router.
7. The authors have released no code. The implementation and credibility of the proposed method are still unclear.

[1] Jun Kong et al., "Accelerating Inference for Pretrained Language Models by UnifiedMulti-Perspective Early Exiting," COLING-2022.
[2] Josef Kalfka et al., "Spying on your neighbors: Fine-grained probing of contextual embeddings for information about surrounding words." ACL-2020.

**Questions:**

See Weaknesses

---

### Official Review · Reviewer_UKZQ · 2024-11-02

**Soundness:** 2
**Presentation:** 2
**Contribution:** 2
**Rating:** 3
**Confidence:** 4

**Summary:**

This paper proposes a fine-grained token-wise pruning method, using a learnable router to adaptively skip less important tokens during inference, reducing computational overhead. The approach includes a search-based sparsity scheduler to construct the router efficiently. Experiments demonstrate that the proposed method achieves state-of-the-art pruning results.

**Strengths:**

1. The story is concise and easy to understand.
2. The method seems concise yet effective.

**Weaknesses:**

Although the approach seems concise yet effective, the lack of many details makes the experiments difficult to find convincing, and some statements are also confusing.
1. In lines 14–16 of the abstract and line 53 of the introduction, I would assume that your approach overcomes previous issues without needing any training. However, training is still required, which is somewhat confusing.
2. Your experiment’s comparisons seem unfair. On one hand, the baselines such as LaCo and LLMPruner, do not require any training but your method needs training. On the other hand, it seems that your compression ratio is the proportion of skipped token hiddenstates, while the baselines pruning methods calculate the ratio based on the parameters removed, which eliminates all computation related to these parameters. Therefore, their actual computational overhead seems to be lower than yours. You could add results of the actual computation cost.
3. Your training details are unclear. The distillation loss and guide loss are not formally defined, making it difficult to understand your training process. Additionally, the size of your training dataset in terms of token count should be provided.
4. The difference between your method and hiddenstate sparsification is unclear. Although lines 137–140 seem to attempt an explanation, your approach essentially appears to be skipping hidden states.
5. You should report on memory usage, as the additional router and intermediate variables introduce new overhead.

**Questions:**

1. Can the scheduler determine the sparsity rate needed for each block based on a given overall sparsity rate?
2. Currently, in Transformer-based LLMs, the output of each layer is calculated like:
attn_output = torch.matmul(attn_weights, value_states)
which computes the hidden states of an entire sentence at once and how you skip the computation for certain tokens? Do you have any code available to share?

---

### Official Review · Reviewer_y7ZL · 2024-11-04

**Soundness:** 3
**Presentation:** 3
**Contribution:** 3
**Rating:** 3
**Confidence:** 4

**Summary:**

This paper proposes a fine-grained token-wise pruning approach (FTP) for large language models (LLMs) to address the issue of high computational overhead during inference. The method involves a learnable router that adaptively skips less important tokens across model blocks. The authors present a search-based sparsity scheduler, a trainable router with specific input factors and losses, and conduct extensive experiments to demonstrate its superiority over existing pruning methods.

**Strengths:**

The proposed FTP method is a novel contribution to the field of LLM pruning. The idea of using a learnable router to adaptively identify and skip less important tokens is intuitive and has the potential to significantly reduce computational cost during inference without sacrificing too much performance.

The authors conduct extensive experiments on various LLMs and benchmarks, comparing their method with several state-of-the-art pruning techniques. The results clearly demonstrate the superiority of the proposed method in terms of accuracy retention and sparsity levels, which adds credibility to the research.

**Weaknesses:**

Incomplete comparison of inference speedup. In Table 6, only the inference speedup ratio of the current model is presented, lacking a comparison with other baseline models under the same conditions. It is not clear whether the FTP's token selection method truly brings significant inference acceleration compared to other methods.

Limited exploration of acceleration ratio across different lengths. There is no comparison of the acceleration ratio with other models at different token lengths. It remains uncertain whether the inference speed improvement of this model becomes less significant as the sentence length increases.

**Questions:**

-In Table 6, why was only the inference speedup of the current model presented without a comparison with other baseline models? How can we be sure that the FTP's token selection method effectively accelerates inference compared to other approaches?

-Have you considered comparing the acceleration ratio of this model with other models at different token lengths? If so, what are the results? Does the inference speed improvement of this model decrease as the sentence length grows?

-Have you considered integrating the Router and the Sparsity Scheduler? If so, what are the potential benefits and challenges of such an integration?

---

### Official Review · Reviewer_A8Z9 · 2024-11-06

**Soundness:** 2
**Presentation:** 2
**Contribution:** 2
**Rating:** 3
**Confidence:** 5

**Summary:**

In this paper the authors propose a method for accelerating inference in fine-tuned Transformer models by learning a routing function that selects whether to calculate updated representations for each token at each layer. The method works by first learning a “static” token router to identify fixed token positions to remove at each layer, with sparsity ($k$ for selecting top $k$ tokens) learned using a genetic algorithm approach. Given those target sparsities, and the static router, dynamic token-level routers are learned for each layer, using a lightweight set of features: position, absolute attention score, relative attention score rank, and layer target sparsity. The dynamic router is learned using standard SGD with a combination of losses incorporating the sparsity constraints, and distillation wrt the original model and static router. There is then an additional finetuning phase for the static sparsities using the learned dynamic router.

In experiments on Llama-2-7B, Llama-2-13B, Llama-3-8B, and Qwen1.5-7B fine-tuned on the Alpaca dataset and evaluated on HellaSwag, MMLU, ARC-easy, ARC-challenge, and WinoGrande, the authors show that the approach can achieve better end-task performance at a comparable compression rate (sparsity level) compared to previous work.

**Strengths:**

- **Important and practical problem:** Improving LLM inference efficiency is an important area of research with many practical applications.
- **Simple and intuitive solution based on analysis:** The solution is intuitive and includes interesting innovations, such as the simple classifier for learning dynamic token routing. It’s based on an initial analysis which demonstrates token redundancy across two model families.

**Weaknesses:**

I really wanted to like this paper, I think it's an innovative approach, but as the paper is currently written the results simply do not support the claims, due to highly insufficient comparison with prior work, particularly as it relates to computational efficiency.

- **Paper could be written more clearly:** In the abstract and introduction, the approach is motivated in part by emphasizing that the approach does not require re-training, however, training is in fact required, to learn the sparsity levels and routing function (About 3 hours of training on a 64GB AMD MI250 GPU). This should be clarified throughout the paper, and it should be made clear early on that this is a task-specific approach for finetuning models. Additionally, various details are missing or poorly located throughout the paper. For example, “token length” is not clearly defined, training time for the dynamic router is included in the main paper but for the static router, and when the static router training is reported in the appendix, it’s not clear whether that includes the final finetuning phase. Batch size is not reported for inference experiments, which is critical for determining the practicality of the approach on accelerator hardware. If the approach only works with batch size 1, then this needs to be clearly addressed and motivated.
- **Related work is insufficiently discussed:** The related work section could more clearly contrast with previous work, and some related work is missing. For example, foundational work on dynamic token-wise computation in Transformers [(Dehghani et al. 2019)](https://arxiv.org/abs/1807.03819), recent work on dynamic token-level routing [(Ainslie et al. 2023)](https://arxiv.org/abs/2303.09752) as well as work on token pruning in BERT models, e.g. [(Goyal et al. 2020)](https://arxiv.org/abs/2001.08950), [(Dai et al. 2020)](https://arxiv.org/abs/2006.03236), and others (these can be used as a starting point), which makes me concerned that other important related works might be missing. The MoD model is highly related, since it also does token-level dynamic routing, but its relation to the proposed work is barely discussed, with no motivation for why different design choices are made in the proposed approach, and the authors don’t compare to this highly related work in experimental results.
- **Insufficient efficiency evaluation compared to previous work, unclear whether approach leads to practical improvements:** Table 6 reports inference efficiency acceleration of the proposed model at different sparsity levels and across different maximum sequence lengths, but does not compare with any previous work in terms of efficiency. Sparsity levels for different approaches are reported in the main results table, but those are insufficient for determining whether there are practical efficiency improvements. Further, while the training efficiency of the method is claimed as a key contribution of the work, there are no experiments comparing training efficiency to prior work.
- **Minimal inference efficiency acceleration:** The maximum inference efficiency acceleration reported in Table 6, for max sequence lengths of 2k, is 1.6x. It’s unclear whether this improvement is achievable with batching; prior work with less dynamic resource allocation does so because it is necessary to leverage the practical computational improvements of batching. This needs to be addressed.

**Questions:**

- How does the tradeoff between inference efficiency and end-task performance compare to baselines? It would be good to see a plot with end-task accuracy on one axis and latency/speedup on the other axis, to see how this approach compares in terms of Pareto optimality to existing approaches.
- In Table 6, what are the actual average token lengths for each set of token length experiments? What batch size is used?
- Is the approach compatible with acceleration using batching?
- Is token length referring to total input/output sequence length, or just input sequence?

Notes:
- Figure 1 could be a bit clearer. I like the color coding for compute vs. skip, and if you continued those colors for the arrows in the digram (for example where you currently use dotted vs. solid arrows), I think this would help. Then, I think you could collapse the two versions of the “token router” zoom in on the right into one, and merge the “token router” component into the circle that determines whether to compute or skip (rather than having additional arrows between the token router and the circle) — the circle seems redundant, since is that not just exactly the token router component?
- Minor grammatical errors throughout the paper, I recommend using Grammarly, an LLM or similar tool to proofread and edit for grammar.
- Fix latex quotation marks on line 741.
- Add information on training time for static router to main paper, and include time for the final finetuning of the static router after training the dynamic router (this would happen naturally with a more detailed comparison of training time to prior work.)

---

### Note · Authors · 2024-11-16

I have read and agree with the venue's withdrawal policy on behalf of myself and my co-authors.